Is it safe to nest near conspicuous neighbours? Spatial patterns in predation risk associated with the density of American Golden-Plover nests

Giroux Marie-Andrée marie.a.giroux@gmail.com 1 2 3 4
Trottier-Paquet Myriam 1 3 4
Bêty Joël 2 3
Lamarre Vincent 1 3
Lecomte Nicolas 1 3 4
1 Canada Research Chair in Polar and Boreal Ecology, Université de Moncton , Moncton , NB , Canada
2 Canada Research Chair on Northern Biodiversity, Université du Québec à Rimouski , Rimouski , QC , Canada
3 Centre d’Études Nordiques, Université du Québec à Rimouski , Rimouski , QC , Canada
4 Quebec Center for Biodiversity Science, Université du Québec à Rimouski , Rimouski , QC , Canada
Kramer Donald
Electronic publication date: 2016 Aug 10
Publication date: 2016
Volume: 4
Electronic Location ID: e2193
Received 2016 Apr 3; Accepted 2016 Jun 8
Copyright: ©2016 Giroux et al.
Copyright year: 2016
Copyright holder: Giroux et al.
License: This is an open access article distributed under the terms of the Creative Commons Attribution License, which permits unrestricted use, distribution, reproduction and adaptation in any medium and for any purpose provided that it is properly attributed. For attribution, the original author(s), title, publication source (PeerJ) and either DOI or URL of the article must be cited.
License URL: https://creativecommons.org/licenses/by/4.0/

Keywords: Arctic, Community, Nest protection, Predation, Shorebirds, Territory, Charadriidae

Funding: W Garfield Weston Foundation Canada Foundation for Innovation Canada Research Chairs Natural Sciences and Engineering Research Council of Canada Université de Moncton Government of Nunavut Indian and Northern Affairs Canada Polar Continental Shelf Project The study was funded by the W Garfield Weston Foundation (fellowship to MAG), the Canada Foundation for Innovation (grant to NL), Canada Research Chairs (Grant to NL), Natural Sciences and Engineering Research Council of Canada (Grant to NL and Environorth scholarship to MAG), Université de Moncton, Government of Nunavut, Indian and Northern Affairs Canada, and Polar Continental Shelf Project. The funders had no role in study design, data collection and analysis, decision to publish, or preparation of the manuscript.

==============================
Predation is one of the main factors explaining nesting mortality in most bird species. Birds can avoid nest predation or reduce predation pressure by breeding at higher latitude, showing anti-predator behaviour, selecting nest sites protected from predators, and nesting in association with protective species. American Golden-Plovers (Pluvialis dominica) defend their territory by using various warning and distraction behaviours displayed at varying levels of intensity (hereafter “conspicuous behaviour”), as well as more aggressive behaviours such as aerial attacks, but only in some populations. Such antipredator behaviour has the potential to repel predators and thus benefit the neighbouring nests by decreasing their predation risk. Yet, conspicuous behaviour could also attract predators by signalling the presence of a nest. To test for the existence of a protective effect associated with the conspicuous antipredator behaviour of American Golden-Plovers, we studied the influence of proximity to plover nests on predation risk of artificial nests on Igloolik Island (Nunavut, Canada) in July 2014. We predicted that the predation risk of artificial nests would decrease with proximity to and density of plover nests. We monitored 18 plover nests and set 35 artificial nests at 30, 50, 100, 200, and 500 m from seven of those plover nests. We found that the predation risk of artificial nests increases with the density of active plover nests. We also found a significant negative effect of the distance to the nearest active protector nest on predation risk of artificial nests. Understanding how the composition and structure of shorebird communities generate spatial patterns in predation risks represents a key step to better understand the importance of these species of conservation concern in tundra food webs.

Predation is one of the main factors causing nesting mortality in most bird species (Ricklefs, 1969), and hence represents a selective force that induced the development of strategies for minimizing nest predation (Smith et al., 2007b). Predation pressure in birds can be reduced as a result of various responses observed at the species and individual levels, such as breeding at higher latitude (McKinnon et al., 2010a), showing anti-predator behaviour (e.g., Simmons, 1952), selecting nest sites protected from predators, and nesting in association with protective species (Haemig, 2001; Quinn & Ueta, 2008). For instance, several studies showed that predation risk increases with the distance to the nest of aggressive or territorial species such as Snowy Owl (Bubo scandiacus; Bêty et al., 2001), Goshawk (Accipiter gentilis; Mönkkönon et al., 2007), Hooded Crow (Corvus corone cornix; Roos, 2002), and Northern Lapwing (Vanellus vanellus; Elliot, 1985).

Shorebirds, particularly the biggest species with colourful plumage and both parents contributing to parental care, are known to display various behaviours to protect their nests (e.g., Drury, 1961; Sordahl, 1981; McCaffery, 1982; Larsen, Sordahl & Byrkjedal, 1996). The American Golden-Plover (Pluvialis dominica, hereafter “plover”) is an example of a shorebird species protecting its nest by using a suite of warning and distraction behaviours displayed at varying levels of intensity, as well as more aggressive behaviour such as aerial attacks in some populations (reviewed in Johnson & Connors, 2010). Such behaviour could decrease predation risks for other species nesting nearby, as shown in another arctic-nesting plover species, the Black-bellied Plover (Pluvialis squatarola; Larsen & Grundetjern, 1997). Yet, this conspicuous behaviour could also attract predators by signalling the presence of a nest (Paulson & Erckmann, 1985). For instance, nesting near a species displaying a conspicuous and aggressive behaviour like the Sabine’s Gull (Xema sabini) can decrease nest survival of Red Phalaropes (Phalaropus fulicarius) in years of high predation pressure (Smith et al., 2007b). The nest density of the species using conspicuous and sometimes aggressive antipredator behaviour could also shape patterns in predation risks for nearby nests. The direction of a density effect would depend on whether the increased nest density better repel (Andersson & Wiklund, 1978) or attract predators (Paulson & Erckmann, 1985; Schmidt & Whelan, 1999; Varela, Danchin & Wagner, 2007). Nests of arctic-breeding birds, including American Golden-Plovers, are mainly depredated by Arctic Foxes (Vulpes lagopus) and avian predators such as Common Raven (Corvus corax), Glaucous Gull (Larus hyperboreus), and Long-Tailed Jaeger (e.g., Bêty et al., 2002; Lecomte et al., 2008). In addition to bird nests, all of these predators forage on the cyclical lemmings, and the foraging intensity of these predators on bird nests increases in lemming crash years (Bêty et al., 2002; McKinnon et al., 2013). Because many arctic-nesting shorebird species are currently experiencing dramatic declines across their range (Morrison et al., 2001; Gratto-Trevor et al., 2011), studying the influence of these species on spatial structures in nest predation risks may shed light on factors favouring nesting associations between arctic-nesting species.

The purpose of the study is to test the hypothesis that the American Golden-Plover can generate spatial structure in predation risks within tundra ecosystems. Based on the results obtained with a congener species, the Black-bellied Plover (Larsen & Grundetjern, 1997), we predict that nest predation risks decrease with (1) the proximity to a plover nest and (2) the density of plover nests. To test the existence of such a protective effect, we conducted an experimental study with artificial nests in a High-Arctic breeding site.

Methods

Study area and species

We conducted the study on Igloolik Island, Nunavut, Canada (69.39°N, 81.55°W; 103 km2) in July 2014 (Fig. 1). There, the tundra landscape is mainly composed of raised beaches with little vegetation, Dryas-lichen slopes, and grass-sedge wet and dry meadows (Forbes et al., 1992). The average annual temperature for the period of 1981–2010 was −12.9 °C with the warmest month (July) averaging 7.6 °C (Environment Canada, 2015). These temperatures and vegetation features correspond to a High-Arctic climate.

Figure 1 Location of the study area on Igloolik Island (Nunavut, Canada, 69.39°N; 81.55°W).

Location of the study area (white circle in (B)) on Igloolik Island (Nunavut, Canada, 69.39N; 81.55W; (A and B)). (C) displays the design of the manipulation conducted in July 2014 with artificial nests placed 30, 50, 100, 200 and 500 m from a focal natural plover nest. Density of active plover nests within a radius of 250 m is shown for the artificial nest placed at 100 m from the focal natural plover nest (two active plover nests in this example).

Igloolik Island is a known breeding site for up to 33 bird species, including shorebirds, waterfowl, and seabirds (Lecomte & Giroux, 2015). Shorebird nest density on the Island was 53.5 nests km−2 for our study (Lecomte & Giroux, 2014, unpublished data). The proximity to the cliffs of Coxe Islands (ca. 15 km away) and to a polynya (ca. 1.5 km away) allow numerous cliff breeders as well as offshore and pelagic species to use our study area as a foraging and resting site. The following nest predators are found on the Island: Arctic Foxes, Ermines (Mustela erminea), Parasitic (Stercorarius parasiticus) and Long-Tailed Jaegers, Glaucous Gulls, and Common Ravens (Ellis & Evans, 1960; Forbes et al., 1992; Lecomte & Giroux, 2014, unpublished data).

Every year since 2013, we conduct live trapping of lemmings (Collared Lemming, Dycrostonyx groenlandicus; brown lemming, Lemmus lemmus) on the study area to determine their abundance. The trapping takes place for five days in late June-early July by setting up 50 Sherman traps at every second intersection (20 m spacing between intersections) of a 200 m × 200 m grid. After the initial set-up, the traps are visited every 4–6 h for 56 h (total of 12 visits). The total number of lemmings captured varies greatly from year to year and ranges from 0–46. In 2014, no lemming was captured, which means that lemming abundance was most likely very low during bird nesting season.

American Golden-Plover nests typically contain four eggs, and both parents incubate the nests during 25–27 days (Johnson & Connors, 2010). On Igloolik Island, success of American Golden-Plover nest varied between 67% and 92% (Lecomte & Giroux, 2014, unpublished data).

Experimental design

To evaluate the protection effect of American Golden-Plovers on their neighbouring nests, 35 artificial nests were placed at different distances from seven plover nests. Each artificial nest consisted of a hen egg placed in an empty nest cup found in the tundra. We used the protocols used in previous studies to determine the ranges of distance from plover nests (Larsen & Grundetjern, 1997; Bêty et al., 2001) and the frequency of visits (Bêty et al., 2001; Nguyen, Abraham & Nol, 2006; Lecomte et al., 2008; Lecomte, Gauthier & Giroux, 2009). The artificial nests were placed along a linear transect at 30, 50, 100, 200, and 500 m from active plover nests (Fig. 1). We selected the orientation of the transect randomly by drawing a number between 0 and 360°. Artificial nests were marked in the same way as the natural nests with two wooden sticks and a blue flag placed 1, 5, and 10 m north of the nest. Artificial nests were deployed with rubber gloves between 12:00 and 18:00 on 7 and 8 July and checked after 1, 2, 4, 6, 8, and 12 days of exposure. The nests were considered depredated when their egg was missing or broken.

In addition to the linear distance (in m) to the associated plover nest, we recorded the following variables: linear distance to the closest active plover nest (in m), density of active plover nests within a radius of 250 m around the artificial nest (Fig. 1; American Golden-Plovers start to respond to predators at 200–300 m, Byrkjedal, 1987), habitat type (wetland or mesic tundra), and vertical nest concealment (estimated for all nests on 20 July). The distance to the closest active nest became different from the linear distance to the associated plover nest when the associated plover nest was depredated because the latter became inactive. Nest density, nesting success, the type of nesting habitat (wetland or mesic tundra) were evaluated following the Arctic Shorebird Demographic Network protocol (SC Brown et al., 2014, unpublished data) within an extensive survey zone of 11.7 km2. Vertical nest concealment corresponded to the percent of the nest obscured when viewed through an ocular tube (PVC pipe, 4 cm diameter × 11 cm length) from 1 m directly above the nest.

Statistical analysis

We modelled the variations in predation risk on artificial nests (response variable) using mixed-effect Cox proportional hazard regression models (library coxme; Therneau, 2012), including the following predictor variables: distance to the associated plover nest (linear and quadratic terms), distance to the closest active plover nest (linear and quadratic terms), density of active plover nests within a radius of 250 m around the artificial nest, habitat type, and vertical concealment. We included the artificial and natural nest identity as random terms. Mixed-effect Cox proportional hazard regression models estimate the relationship between Kaplan–Meier survival estimates and the response variables. The exponent of the parameter estimate for each response variable provides the estimate of the hazard ratio, which corresponds to the hazard risk (or predation risk in our study) relative to a baseline measure of risk.

We used a model selection approach (Burnham & Anderson, 2002) to identify the combination of these variables that best described variations in the predation risk of artificial nests. We compared 24 biologically plausible, candidate models, including up to four of the predictors described above in a single model (see Table S1 for the full list of models). We did not include predictors displaying multicollinearity (r < 0.70) in the same model (Dormann et al., 2013). We identified the combination of predictors that best described variations in predation risk using the corrected Akaike Information Criterion (AICc) for small sample size (Burnham & Anderson, 2002) estimated from the library AICcmodavg (Mazerolle, 2015). Models with ΔAICc < 2 from the top model were considered competitive (Burnham & Anderson, 2002). Finally, we used the survfit function (library survival, Therneau, 2015) to create survival probability curves using the Kaplan–Meier survival estimates of Cox models.

We tested the assumption of the Cox models that the hazard function does not change over time for each covariate by regressing the Schoenfield residuals across time (Hess, 1995). A significant non-zero slope indicates a violation of the assumption. We confirmed that the assumption was respected for each predictor variable through visual inspection of the regression of the Schoenfield residuals against time confirmed, and also for each model (cox.zph function, library survival; Therneau, 2015). We performed all statistical analyses in R 3.2.3 (R Development Core Team, 2015).

The experiment and field protocols were approved by the Université de Moncton Animal Care Committee (permit # 14-05), by the Department of Environment—Government of Nunavut (permit # WL-2014-039), and by the Canadian Wildlife Service (permits #NUN-SCI-14-04).

Results

A total of 18 American Golden-Plover nests was found in our extensive search area of 11.7 km2 (1.5 plover nests km−2). All of these nests were active when we started the experiment on 7 and 8 July. We therefore used more than a third of all available nests (seven vs. 18 nests) to run the experiment.

Predation risk and density of active plover nests

The model that best explained variation in predation risk on artificial nests included the density of active plover nests within a radius of 250 m around the artificial nest (Table S1). The Cox proportional hazard mixed-effects regression model indicated that the predation risk increased by 1.4-fold (coefficient = 0.87, SE = 0.24, P = 0.0003, hazard ratio = 2.4; Fig. 2), when we observed one active plover nest within the 250-m radius around the artificial nest. The increase was 2.4-fold (coefficient = 1.22, SE = 0.37, P = 0.001, hazard ratio = 3.4; Fig. 2) when there were two active plover nests within the 250-m radius. The second most parsimonious model (ΔAICc = 1.99) included the effect of habitat type in addition to the density of active plover nests (Table S1). However, the effect of habitat type on predation risk was not significant (coefficient = 0.09, SE = 0.24, hazard ratio = 1.10, P = 0.7).

Figure 2 Kaplan–Meier survival probabilities over 12 exposure days for artificial nests with varying active plover nest density (zero, one or two active nest[s] with a radius of 250 m around the artificial nest) on Igloolik Island (Nunavut, Canada) during the summer of 2014.

Each data point on the curve represents the Kaplan–Meier survival estimate at time t (±SE), which provides the probability that a nest will survive past time t.

Predation risk and distance to the nearest active plover nest

To confirm the direction of the results obtained through the best fitting model shown above, we also report the results of the model including the distance to the nearest active plover nest, although this model had a ΔAICc > 2 (ΔAICc = 2.36; Table S1). This model showed that predation risk of artificial nests decreased by 20% for each additional 100 m further away from an active plover nest (coeff = −0.21, SE = 0.06, P = 0.0003, hazard ratio = 0.81).

Discussion

We showed that predation risk on artificial nests increased with the density of active plover nests and decreased with the distance to the nearest active plover nest during a year of low lemming abundance. Contrary to our predictions, these results do not support the existence of a protective effect of nesting plovers on nearby nests.

Spatial variation in predation risk

We predicted that predation risk would decrease with the distance to plover nests, assuming that the antipredator of the American Golden-Plovers could repel predators. Several studies showed that predation risk increases with the distance to the nest of an aggressive or territorial species (Bêty et al., 2001; Mönkkönon et al., 2007; Roos, 2002; Larsen & Grundetjern, 1997; Elliot, 1985). For instance, Larsen & Grundetjern (1997) showed a decrease in predation risk of natural nests with the distance from Black-bellied Plover nests. Yet, some studies have shown that there are no effects (Larsen & Grundetjern, 1997: Pacific Golden-Plovers [Pluvialis fulva]) or even some disadvantages to nesting around a species using conspicuous antipredator behaviour, especially during years of higher nest predation rates (Smith et al., 2007b). For instance, nesting near an aggressive species like the Sabine’s Gull (Xema sabini) increased nest survival of Red Phalaropes (Phalaropus fulicarius) but only in years when nest predation rates were generally low due to high lemming abundance; when lemming abundance decreased, nesting near Sabine’s Gulls induced negative effects on phalarope nest survival (Smith et al., 2007b). These results suggest that conspicuous behaviour may attract shared predators in years when the abundance of the main prey is low.

We also predicted that predation risk would decrease with the density of plover nests. We rather observed that predation risks increase with the density of plover nests. This is not in line with what has been found by Andersson & Wiklund (1978), who showed that predation risk on artificial nests was higher in absence than in presence of a fieldfare pair and nearby a solitary pair than a nesting colony. They attributed their results to their efficient solitary and communal aggressive behaviour towards predators. Our results rather indicate that an increased density of American Golden-Plover nests may attract predators. The presence of species using conspicuous behaviour can attract predators (Smith et al., 2007b), and this can increase with the nest density of the conspicuous species owing to the increased detectability of potential prey by predators (Schmidt & Whelan, 1999; Varela, Danchin & Wagner, 2007).

Consistent with the results of Smith, Gilchrist & Smith (2007a), we found no habitat effect on the survival of the artificial nests. Powell (2001) reported that habitat characteristics were not a good predictor for the nest survival of snowy plover. Yet, some studies provide evidence of a spatial heterogeneity in predator activity (Schmidt, Ostfeld & Smyth, 2006). For instance, Lecomte et al. (2008) reported a higher predation risk in the mesic tundra compared to arctic wetlands because the physical structure of wetlands reduced the mobility of foxes, the main arctic predator for breeding birds.

Prey and predator behaviours

The differences between the antipredator behaviour of the American Golden-Plover and that of the Black-bellied Plover could help explain why our results differ from those of Larsen & Grundetjern (1997), who observed a reduced predation risk around Black-bellied Plover. In addition to alarming, mobbing and distraction displays, Black-bellied Plovers attacked 50% of predators entering an area of 200-m radius surrounding their nests, respectively (Larsen & Grundetjern, 1997). The American Golden-Plover uses various levels of intensity of alarming, mobbing and distraction displays when the predator is within 200–300 m from its nest (Drury, 1961; Gochfeld, 1984; Byrkjedal, 1987; Byrkjedal & Thompson, 1998). That species can also adopt more aggressive behaviours such as aerial attacks in some populations (reviewed in Johnson & Connors, 2010). Yet, aerial attacks by the American Golden-Plovers are considered rare and would occur only when predators are small (Sordahl, 1981; McCaffery, 1982; Paulson & Erckmann, 1985). Because the American Golden-Plover can adopt various behaviour to deter different predator species (Sordahl, 1981; McCaffery, 1982; Paulson & Erckmann, 1985), further studies should aim at determining how nest predation by different species varies with distance from and density of American Golden-Plover nests.

When lemmings are scarce, arctic foxes increase their foraging intensity on bird nests (Bêty et al., 2002; McKinnon et al., 2013), and conspicuous antipredator behaviours could signal the presence of a nest to a predator in search of alternative prey (Smith et al., 2007b). In this study, we did not compare the influence of American Golden-Plover nests on predation risk between years of high and low lemming abundance. Thus, we cannot exclude that low lemming abundance could contribute to explaining the increase in predation risk with the density of and proximity from plover nests, but this remains to be tested.

Estimating predation risk using artificial nests

Artificial nests have the advantage of providing a standardized measure of predation risks. Yet, predation rates on artificial nests differ from that of real nests and, therefore, they should not be used to infer predation pressure on natural nests (Moore & Robinson, 2004; McKinnon et al., 2010b). In our study, we used artificial nests to provide a controlled measure of relative predation risk at various distances from plover nests, not to infer real nest success. Success of natural nests is not only determined by predation risk, but by a combination of factors such as nest defence behaviour (Kis, Liker & Szekely, 2000), parental care (Smith, Gilchrist & Smith, 2007a), incubation duration (Schamel & Tracy, 1987), nest site selection (Martin, 1998), and frequency of incubation recesses (Martin, Scott & Menge, 2000). Here, artificial nests allowed us to control for such sources of heterogeneity to estimate the influence of plover nests on nest predation risk.

Conclusion

In conclusion, the artificial nests experiment conducted on Igloolik Island during a year of low lemming abundance, does not support the existence of a protective effect of plover nests on nearby nests, and in fact implies that it is considerably less safe to nest near plovers. Our results bring new perspective on how the spatial distribution and composition of shorebird communities may influence breeding success of arctic-nesting birds. Understanding how the composition and structure of shorebird communities generate spatial patterns in predation risks represents a key step to better understand the importance of these species of conservation concern (Morrison et al., 2001; Gratto-Trevor et al., 2011) in tundra food webs.

Supplemental Information

Data S1 Dataset of nest predation on artificial nests around American Golden-Plover nests

Click here for additional data file.

Table S1 Selection of models explaining variations in the risk of predation of artificial nests over 12 exposure days on Igloolik Island (Nu, Canada) during the summer of 2014 (n = 24)

We compared models including up to four of the following predictors: distance in metres to the associated plover nest (Distance), distance in metres to the closest active plover nest (Active Distance), density of active American Golden-Plover nests within a radius of 250 m around the artificial nest (Active Density), habitat type, i.e. wetland or mesic tundra (Moisture) and vertical concealment (Concealment). We also tested for quadratic effects of Distance and Active Distance. We report Akaike’s information criterion corrected for small sample size (AICc), difference in AICc relative to the model with the lowest AIC (ΔAICc), as well as the AICc weight (ωAICc). Models are ranked by their AICc values and the best-fitting models (ΔAICc >2) are shown in bold.

Click here for additional data file.

We thank M-C Frenette, Amanda Taqtaq, and Mike Qrunnut for their valuable assistance in the field as well as the Government of Nunavut for its logistical support. We are grateful to D Kramer and two anonymous reviewers for insightful comments on earlier versions of the manuscript.

Additional Information and Declarations

Competing Interests

Author Contributions

Animal Ethics

Ethics

Field Study Permissions

Data Availability

The authors declare there are no competing interests.

Marie-Andrée Giroux conceived and designed the experiments, performed the experiments, analyzed the data, wrote the paper, prepared figures and/or tables, reviewed drafts of the paper.

Myriam Trottier-Paquet conceived and designed the experiments, performed the experiments, analyzed the data, wrote the paper, reviewed drafts of the paper.

Joël Bêty contributed reagents/materials/analysis tools, reviewed drafts of the paper.

Vincent Lamarre analyzed the data, prepared figures and/or tables, reviewed drafts of the paper.

Nicolas Lecomte conceived and designed the experiments, performed the experiments, analyzed the data, contributed reagents/materials/analysis tools, wrote the paper, reviewed drafts of the paper.

The following information was supplied relating to ethical approvals (i.e., approving body and any reference numbers):

The experiment and field protocols were approved by the Université de Moncton Animal Care Committee (permit # 14-05), by the Department of Environment –Government of Nunavut (permit # WL-2014-039), and by the Canadian Wildlife Service (permits #NUN-SCI-14-04).

The following information was supplied relating to ethical approvals (i.e., approving body and any reference numbers):

The experiment and field protocols were approved by the Université de Moncton Animal Care Committee (permit # 14-05).

The following information was supplied relating to field study approvals (i.e., approving body and any reference numbers):

The experiment and field protocols were approved by the Department of Environment –Government of Nunavut (permit # WL-2014-039), and by the Canadian Wildlife Service (permits #NUN-SCI-14-04).

The following information was supplied regarding data availability:

The raw data has been supplied as Data S1.

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
