# Peer review of "Is it safe to nest near conspicuous neighbours? Spatial patterns in predation risk associated with the density of American Golden-Plover nests"

_PeerJ, doi:10.7717/peerj.2193_

## Round 0.1 · original submission · Minor Revisions

· Academic Editor

Minor Revisions

This study tests the hypothesis that the risk of nest predation is lowered in the vicinity of American golden plover nests due to active defense by plovers. The authors set out artificial nests at different distances from real nests and predicted that survival of artificial nests would decrease with distance from real nests. For each artificial nest, they also measured several variables associated with proximity of natural nests and habitat. The best model showed an increase in risk associated with an increase in plover nests in the vicinity. Although not included in the set of best models, risk also increased with proximity to plover nests. These results were the opposite of the predictions. The authors do not attempt to explain the pattern beyond noting that nest defense might attract predators, especially in years such as the study year in which lemming prey are scarce.

Both reviewers were generally positive about the design, analysis and presentation of the study. Reviewer 1 recommended minor revisions, but Reviewer 2 recommended rejection on the grounds that the study was carried out in only one year and would be more suitable as a short communication in an ornithological journal. Because PeerJ does not use a criterion of ‘importance’ or ‘impact’ for the publication of a scientifically valid study and because ecologically studies carried out in a single field season are widely considered as acceptable, despite the recognition that conditions change among years, my decision is that the manuscript is acceptable pending minor revision.

Regarding content, my comments below re- emphasize some of the points made by the reviewers and provide some additional suggestions of my own. I also note a number of minor grammatical and stylistic errors corrections in addition to the detailed suggestions generously provided by Reviewer 2. My specific comments below can be treated as a third review; i.e., either modify the manuscript appropriately or provide a detailed explanation for why you chose not to do so.

Specific comments

Title: I suggest replacing ‘bold’ with an alternative word recognized in the nest defense literature. I initially assumed your study concerned animal temperament, i.e. bold vs. shy, and other readers may respond similarly. (This also applies to L167).

Abstract:
L26. Provide full common and scientific names in the abstract.
L27-29. This is a confusing sentence. What about something like ‘We predicted that the predation risk of artificial nests would decrease with proximity to and density of plover nests’?
L30. ‘found’ is better than ‘showed’ here
L35 represents (sing.)

Keywords: Consider adding the family name of the study species

Introduction:
The Introduction lacks several points important for understanding the rest of the manuscript.
• First, the study species is presented as an active defender of nests, but in the Discussion, you indicate the possibility that it is a weak defender. I suggest that you provide a more complete indication of what is known about the nest defense and relate it to other species in which similar studies have been carried out.
• Second, you justify the distance prediction but not the density prediction. Obviously, the two are outcomes from the same underlying process, but you need to explain why both measures are useful. If you do this carefully, you might be able to help readers anticipate your finding that experimentally controlled distance was a weaker predictor than an observational, controlled variable.
• Third, although you mention the alternative that nest defense could attract predators in the Discussion, it would be appropriate to mention, at least briefly, in the Introduction that other patterns have been found (no effect on risk, risk increasing with proximity) and evidence or speculation regarding the processes involved.
• Third, you do not mention the importance of lemming density, so your trapping methods come as a surprise. (I have additional suggestions on this topic below.)
• Finally, if you think it worthwhile to note the possible effect of different predator species as suggested by Rev. 2, this should be raised earlier in the Introduction, not with the predictions as you might infer from the line reference.
L53. Grey Plover (caps). I have always found it odd that ornithological journals capitalize the common names of birds but not those of other taxa. However, PeerJ is quite flexible in its style, so this is ok, if it is your preference and you are careful to be consistent.
L61. ‘main’ purpose implies that there are other purposes. If you are not going to list them, delete ‘main’.

Methods:
Study area and species
I suggest that you include information that 2014 was a low lemming year in your study area information. As Rev. 1 notes, your evidence needs to include not only that you caught no lemmings in the study year, but comparative evidence of numbers caught using similar methods in other years. Because your data are only from this year, this information might be more clearly presented here and removed from subsequent Methods and Results.
Experimental design
I strongly agree with Rev. 1 that you need to provide more details concerning your artificial nests, including the number and type of eggs, how a potential artificial nest site was defined, and whether the experimental placements were in a linear arrangement. A reader should be able to repeat your study based on information you provide.
Also, why a radius of 270 m for nest density was selected.

L75. waterfowl (no s)
L76, 143 and elsewhere. nests.km-2 (nests not nest)
L77 ca. (abbreviation needs period)
L89. 'blue flag placed 1, 5, and 10 m north of the nest'
L98. This sentence is not clear. What is the random allocation referred to? Does this imply that sometimes plovers nested closer to your locations after the experiment began or that you discovered plover nests you were previously not aware of or do you mean something else?
L103, 151 and elsewhere. Separate number and units with 1 space.
L105. live trapping of lemmings. (Did you already provide the scientific and common names?)

Results:
Remove lemming results if presented in the study area section of Methods
L142. A total . . . was found (sing.)
L145, 153. nests, not nest
L148-152. Long, complex sentence. I suggest dividing into 2 sentences with the condition adjacent to the pattern, e.g., 'The Cox proportional hazard mixed-effects regression model indicated that the predation risk increased by 1.4-fold (coefficient = 0.87, SE = 0.24, P = 0.0003, hazard ratio = 2.4; Fig. 2) when we observed one active plover nest within the 270 m-radius around the artificial nest. The increase was 2.4-fold (coefficient = 1.22, SE = 0.37, P = 0.001, hazard ratio = 3.4; Fig. 2) when there were two active plover nests within the 270 m-radius.'

Discussion:
I agree with Rev. 1 that the Discussion seems a bit long. In general, I think that the solution is tighter organization rather than leaving out any literature or ideas. For example, L189-193 overlaps L163-166, and L166-168 adds little. A more concise discussion might be organized as follows:
• Overview of your findings
• Strength/statistical support for the patterns, relationship between density and distance effects, why support was stronger for density than distance (not currently discussed), and any qualifications due to the artificial nest method. [Note that L193-197 seems to imply that the density effect is more expected from the literature than the distance effect, but you have not made it clear how density could increase without an effect on distance (measurement scales?)].
• Literature context, expanding information provided in the Introduction, including factors such as alternative prey abundance and characteristics of the nest defense and of predators that might influence the pattern.
• Possibility that your unexpected pattern is related to low lemming abundance (or to weaker nest defense than expected, if this is what you are trying to argue).
• Implications for conservation and understanding the spatial distribution of nesting and nest predation.

L164 increased, decreased
L165-168 I do not recommend changing these sentences to past tense as recommended by Rev. 2. Unlike the previous sentence, they refer the current situation (present tense).
L169. Remove this subhead because it is the only one so does not aid the organization. If kept, it should be 'variation', not 'variations'.
L176 rates were
L184 during the summer of 2014 could contribute to explaining
L193 I think that the paragraph comparing your results to Larsen & G 1997 should begin with this sentence and not be split at L201. In general, this section of the discussion is very confusing in its switching between species and not be consistently clear about which species is being discussed. It needs a careful rewrite.
L197-201. This sentence is confusing, in part because 'this study' implies the study published here, although I think you mean the previously mentioned article by Larsen & G. There is also an extra 'that' and the verb should be singular. If I understand correctly, it might be clear to write something like: 'Although that study was performed with natural rather than artificial nests, it still provides an interesting point of comparison to better interpret our results.' [The attempt to include the problem of artificial nests made the sentence too complex, so should be addressed elsewhere.]
L202. Also confusing. Did L & G find species difference? You did not tell us that.
L205. No previous or subsequent mention of Black-bellied Plover. Where do they fit in?
L212 attacking
L215 no habitat effect on the survival
L219 because the physical structure of wetlands reduced the mobility of foxes
L229. Not clear how the next two sentences, especially the reference to feeding associations, provide examples for the preceding statement
L230-233 italics for scientific names
L236 'meaningful ecological statements' is extremely vague
L237 'predation risk in arctic-nesting birds' is much too general a statement; you studied just one species in one site in one year.

Reviewer 1 ·

Basic reporting

Basic reporting is mostly fine. The Discussion is in my view too long relative to the contribution.

More detail on the artificial nests is required. You reference methods used in several other studies but these are so pertinent that they need to be reported here. For example, what sort of eggs did you use in the artificial nests?

You assert repeatedly that it was a low lemming year. I think that rather than details of the trapping method (zero lemmings caught) it would be more informative to report the numbers caught in other years to demonstrate your evidence for this.

What can you tell us about who did the depredation?

Experimental design

An excellent experimental design and compelling statistical analysis.

Validity of the findings

The basic finding is valid, but too much emphasis is placed on the 'low lemming year' interpretation. For example, the Discussion opens on this point, with the implication that results would be different in a high lemming year. 'Years' were not replicated, and any number of things might have been different, other than lemming abundance. This qualifier ought to appear later rather than being marched in front as if you had demonstrated it. It's a possible explanation (viz lines 177 - 178) and should be treated as speculation. Some of the language presents this as though it is a foregone conclusion that this explains your result. For example line 245 is

In conclusion, the artificial nests experiment conducted on Igloolik Island do not support the
246 existence of a protective effect of plover nests on nearby nests, and rather show that it might not
247 be safe to nest near a bold neighbour during years of low abundance of the main prey.

whereas it would be more accurate to say

In conclusion, the artificial nest experiment, conducted on Igloolik Island during a year of low abundance, does not support the existence of a protective effect of plover nests on nearby nests, and in fact implies that it is considerably less safe to nest near plovers.

Also, since this is what Smith (2007b - line 178) reported earlier, why is your result 'surprising' (line 30)?

Reviewer 2 ·

Basic reporting

The paper was generally well-written, but its quality began to deteriorate near the end. English is not the first language for the authors and therefore I assume that serves as an explanation. There were a number of places in the text where I made suggestions for improvements, but I'd recommend a thorough editing by a native English speaker before it goes out again for review.

Experimental design

The design was simple, and in one sense was sufficient for testing the hypothesis; they were interested in determining whether nesting near a species that defends its nest against predators reduces the likelihood that eggs in an artificial nest near the focal nest would have a lower probability of being destroyed. The design for addressing this question was fine (locating artificial nests at varying distances from focal nests), but the study overall was very weak because it was a single-year study with a small sample size (only 7 focal nests of American Golden Plovers). The study also failed, in my opinion, to offer evidence to justify the hypothesis that they posed, and they failed to acknowledge in the introduction the equally plausible hypothesis that nesting near another bird might be disadvantageous because predators might be drawn to areas of high nest density.

Validity of the findings

The statistical analysis was fine, but the sample size was quite small. I am always leery of ecological studies based on data collected in a single year, and this study represents one such example. It may have been a very unusual year (no alternative mammalian prey for predators) and by the authors own admission this may have had a strong influence on the results. The results are interesting, and possibly worth publishing. However, the venue would seem more as a short communication in an ornithological journal.

Additional comments

General comments:

The paper by Giroux et al. (Is it safe to nest near bold neighbours? Spatial patterns in predation risk associated with the density of American Golden-Plover nest) addresses an interesting question: does nesting near a neighbor that aggressively defends its nest against predators have spillover effects that benefit individuals nesting near them? The authors used American Golden Plovers (AGP) as their “aggressive” species, and time to failure for artificial nests placed at varying distances from focal AGP nests. The design of the experiment was fine, but two issues are of concern that have bearing on the results. One is the limited sample size (N = 7) and the other is that the study represents data from a single year. These are not necessarily fatal concerns, but they do have to be considered. Nest success in birds can vary greatly from year-to- year and spatially within years, so I am always concerned when I see a single year study with small sample size. The authors themselves acknowledge the potential issue with a single year study when they mentioned that their results might have been influenced by the fact that 2014, the year of the study, had virtually no lemmings available. The authors suggest that as a consequence of the lack of lemmings as a food source, nesting near AGP may have increased the likelihood that an artificial nest was discovered because predators were drawn to AGP nests. Maybe, but without data from other years we’ll never know (but it’s not a bad hypothesis).
One of my major concerns was the failure in the introduction to posit the obvious alternative hypothesis that was in fact consistent with the results obtained, namely that predators may respond positively to nest density and therefore nesting near other nests would be disadvantageous. The entire premise for their stated hypothesis was presumably that AGP defend their nests aggressively and therefore individuals nesting near them might benefit from the AGP’s tendency to drive predators from the area. Not unreasonable, except that the authors (1) fail to provide any evidence that AGPs actually do defend their nests aggressively, and instead, (2) it is stated in the discussion (line 192) that “American Golden Plover crouches, leaves the nest and sometimes approaches the predator, but never attacks (Byrkjedal and Thompson 1998).” So why did you expect that there would be spillover effects that from nesting near AGP nests?
On a more mundane (but important) level, I also felt that more information was needed on the natural history of the species, and that the methods lacked some important details. A major shortcoming was the treatment of the lemmings. The questions of lemmings as alternative prey, the abundance of which might affect the predator’s behavior, was were never mentioned in the introduction and therefore the sudden appearance of this in the methods was a surprise. We didn’t learn why lemmings were included until the discussion. I concur that they might influence predator behavior in a way that could affect the results, but with only a single year of data we don’t know the answer. Specific line-by-line comments are given below.

Specific comments:
1. Abstract, line 22: what about selection of nest site? Seems like this might even be the most important factor.
2. Abstract, line 23: Need to identify the specific species..."plover" seems to general.
3. Line 37: I'd say "causing" rather than "explaining"
4. Lines 39-42: You seem to be conflating species-level and individual-level responses. Individuals do not "choose" to breed at different latitudes...that is a species-level phenomenon. On the other hand, showing anti-predator beahviour could be both a species- and individual-level response, as could nesting in association with protective species. Moreover, you never mention selection of nest sites, which may be the most important response to predation risk.
5. Line 53: “that actively defends” instead of “actively defending”
6. Line 54: “depredated” instead of “predated”. “Predated” means to come or happen before.
7. Lines 55-56: With respect to common names, you used lower case for the mammal and upper case for the birds. I know that the AOU uses upper case for common names, but be consistent and follow PeerJ's formatting for common names.
8. Lines 59-60: This justification has a very shallow ring to it….be honest and simply say something to the effect that the study might shed light on factors that may favor close associations between different species.
9. Lines 62-64: Seems like the predictions might change with the nature of the predator. Is it a predator that hunts using olfactory or visual cues? What about its size relative to the prey species?
10. Lines 63: There is an obvious alternative and it is that it pays to be a solitary nester and avoid high density areas that might attract predators. Why isn't this presented as an equally plausible hypothesis? I’ll comment on this in greater detail when I get to the Discussion.
11. Line 79: Are any of these known to depredate American Golden Plover (AGP) nests? How about providing some information on nest success of AGPs in this area? Some additional background on their biology would be helpful. For instance, what is typical clutch size (I assume 4 eggs)? How long is incubation? Do both parents participate in incubation?
12. Line 84: “placed” instead of “disposed”.
13. Line 89: We need details on the artificial nests. How were they constructed? Were eggs placed in them? How many? If more than one egg, were eggs placed in them all at once or on sequential days?
14. Line 95: Why was a radius of 270 m chosen? What is the justification?
15. Line 103: When was coverage above the nest measured? At what point in the experiment? All nests were set up on either 7 or 8 July…was cover measured at that point or when the nest failed?
16. Line 104: This was totally unexpected? You never mentioned anything about lemmings in the introduction and at this stage I could see them as being possibly relevant for 1 or 2 reasons, and they would predict different results. You need to address, in the introduction, the potential influence of lemmings on the outcome of your experiment to justify this section.
17. Line 125: You have a tendency throughout to use the present tense when the past tense should be used. This should be “identified” rather than “identify”.
18. Line 142: Were the 18 nests all active at the same time? Did they all contain eggs on 7 and 8 July when the experiment began?
19. Lines 163-168: This should be written in the past tense. You began correctly with “showed” but then every verb after that is in the present tense.
20. Line 166: Where is the evidence that active nest defense by an AGP has a protective effect on its own nest?
22. Line 166: With respect to the "various factors"...you focused on proximity and density to American Golden Plover nests, but what about TOTAL nest density? What about other species nesting in the area?
23. Line 167 “Unexpected results”: I don't see them as unexpected...it conforms well to the alternative hypothesis that nesting at low density is advantageous.
24. Line 179: What qualifies as "conspicuous behaviour"...does this mean overt and aggressive nest defense behaviour?
25. Lines 186-187: I think you mean to say "may have attracted rather than repelled predators"
26. Line 188: Instead of “lemming abundance” you should say “years of high and low lemming abundance."
27. Line 189: It could be argued that these are pretty much one and the same.
28. Lines 193-195: If you had this reference available to you, why would you not pose this as your hypothesis from the start? It should have at least been treated as an alternative hypothesis from the start and described in the introduction.
29. Line 204: "an area of" instead of “a”
30. Line 206: "help explain" instead of “contribute explaining”
31. Lines 209-211: Your original hypothesis is completely inconsistent with this type of behaviour. Consequently, I'd have to say you have no grounds for positing the hypothesis that you did. You should have posed the alternative hypothesis, which in fact conforms to the results that you present.
32. Line 227: I keep going back to this, but you should also mention nest placement.
33. Lines 230-233: need to italicize scientific names.
34. Lines 366-367: Use lower case for words in title, and remove “The” from “The Condor”.

---

## Round 0.2 · accepted · Accept

· Academic Editor

Accept

The manuscript has been appropriately revised and was accompanied by a carefully prepared rebuttal statement facilitating my review. I consider it now ready for publication. Two minor redundancies noted to PeerJ staff can be addressed during the publication process.

L108-109. Redundant word 'nest'. I suggest removing first 'nest' and changing 'Golden-Plover' to 'Golden-Plovers'.

The permit statement on L135-137 is repeated on L167-169.